# Associations of Infant Feeding, Sleep, and Weight Gain with the Toddler Gut Microbiome

**DOI:** 10.3390/microorganisms12030549

**Published:** 2024-03-09

**Authors:** Magdalena Olson, Samantha Toffoli, Kiley B. Vander Wyst, Fang Zhou, Elizabeth Reifsnider, Megan E. Petrov, Corrie M. Whisner

**Affiliations:** 1College of Health Solutions, Arizona State University, Phoenix, AZ 85004, USA; mamezqu3@asu.edu (M.O.); samntoffoli@gmail.com (S.T.); kbvanderwyst@gmail.com (K.B.V.W.); fzhou42@asu.edu (F.Z.); 2Center for Health Through Microbiomes, The Biodesign Institute, Arizona State University, Tempe, AZ 85281, USA; 3Edson College of Nursing and Health Innovation, Arizona State University, Phoenix, AZ 85004, USA; elizabeth.reifsnider@asu.edu (E.R.); megan.petrov@asu.edu (M.E.P.)

**Keywords:** 3-year-old gut microbiome, infant sleep, early life behaviors

## Abstract

This study examines how feeding, sleep, and growth during infancy impact the gut microbiome (GM) in toddlers. The research was conducted on toddlers (*n* = 36), born to Latina women of low-income with obesity. Their mothers completed retrospective feeding and sleeping questionnaires at 1, 6, and 12 months; at 36 months, fecal samples were collected. Sequencing of the 16S rRNA gene (V4 region) revealed that breastfeeding for at least 1 month and the introduction of solids before 6 months differentiated the GM in toddlerhood (Bray–Curtis, pseudo-F = 1.805, *p* = 0.018, and pseudo-F = 1.651, *p* = 0.044, respectively). Sleep had an effect across time; at 1 and 6 months of age, a lower proportion of nighttime sleep (relative to 24 h total sleep) was associated with a richer GM at three years of age (Shannon H = 4.395, *p* = 0.036 and OTU H = 5.559, *p* = 0.018, respectively). Toddlers experiencing rapid weight gain from birth to 6 months had lower phylogenetic diversity (Faith PD H = 3.633, *p* = 0.057). These findings suggest that early life nutrition, sleeping patterns, and growth rate in infancy may influence the GM composition. Further verification of these results with objective sleep data and a larger sample is needed.

## 1. Introduction

Childhood obesity often persists into adulthood and may increase the risk of the early onset of metabolic conditions [1]. Moreover, predictors of childhood obesity are high body mass index (BMI) and rapid weight gain (RWG) in infancy [2]. Breastfeeding status and the timing of solid food introduction have been associated with long-term metabolic health [3,4]. Similarly, sufficient and high-quality sleep in infancy is important for optimal cognitive development [5], physical growth [6,7,8,9], and future health trajectories [10].

A possible mechanism by which these early practices and growth trajectories may impact long-term health and development is via the gut microbiome (GM). The infant GM differs from that of an adult in various aspects; for instance, the infant GM exhibits lower species richness and is considered to be ‘milk-adapted’; rich in *Bifidobacterium* and lacking microbes capable of digesting complex carbohydrates such as *Bacteroidaceae, Lachnospiriceae,* and *Ruminococcaceae* [11,12]. Moreover, the infant GM is highly variable among individuals and across time points due to differences in transient and colonizing bacteria, making it a challenging population to study [12]. The GM matures during the first three years of life by acquiring new species of microbes until its composition resembles that of an adult and remains resilient to changes into later adulthood, supporting immune and metabolic functions [13,14,15,16]. Previous studies have indicated that an individual’s genetic makeup, their unique behaviors, and their surrounding environment influence the GM composition and development [12,14,17]. However, the ways in which infancy practices and experiences contribute to shaping the toddler GM composition during this critical developmental window have yet to be fully understood.

Nonetheless, recent studies are beginning to shed light on this complex interplay, exploring how specific infancy practices, such as feeding and sleep patterns, correlate with the development and composition of the GM. For example, the unique nutrients found in breastmilk, along with the microbial transfer associated with breastfeeding, could shape the infants’ GM [18]. Indeed, it has been shown that breastfed infants exhibit a different GM composition than formula-fed infants [11]. However, it is unclear if the differentiation related to breastfeeding status and duration attenuates as the infant matures into toddlerhood [19,20,21] or has a long-lasting effect [22,23]. Variation among studies suggests other factors such as breastfeeding exclusivity (no formula), formula supplementation, and timing of solid food introduction may affect the feeding mode-GM relationship. In parallel, sleeping patterns could influence the GM via hormone secretion, circadian rhythms, or eating patterns [24]. Self-reported sleep quality, sleep efficiency, and total sleep time (TST) have been positively correlated with GM diversity in adults [25,26]. Although the evidence of the sleep-GM relationship in adults continues to grow [24], little is known about the infancy period. One study found three-month-olds who slept more during the daytime had lower alpha diversity, indicating reduced microbial richness, but this differentiation was not observed in 6- or 12-month-olds [27]. Finally, we know the GM of obese individuals has a higher capability of energy harvest than the GM of lean individuals [28], but the influence of RWG on the GM remains less studied.

Feeding practices, sleep patterns, and weight gain are all modifiable; with a greater understanding of their contributions to GM development, scientists and clinicians may be able to develop effective interventions that promote GM health and prevent metabolic disease. In this observational study, we hypothesized that infant growth and behaviors would be prospectively associated with toddler GM diversity and composition.

## 2. Materials and Methods

This study was a secondary analysis of data from a randomized clinical trial (NIDDK 5R01DK096488-03) assessing an infant feeding and development education program’s efficacy in preventing overweight [29]. The institutional review board at Arizona State University and the state and city health departments approved the informed consent and research protocol.

Pregnant Latina women with low-income and obesity (*n* = 126) were recruited from the Special Supplemental Nutrition Program for Women, Infants, and Children (WIC) programs in Houston, TX, USA. Latina women are recognized as a particularly vulnerable group due to their increased risk of obesity, which also extends to their offspring [30,31]. At baseline, maternal demographics, prenatal BMI, parity, delivery mode, infant sex, and birth anthropometrics were collected via self-report.

At 1, 6, and 12 months of the infants’ age, mothers retrospectively reported their infant’s feeding practices (Infant Feeding Observation Questionnaire) [29,32] and sleep patterns (Brief Infant Sleep Questionnaire) [33] and study staff measured infant weight and length using standardized equipment. Feeding variables of interest were breastfeeding status, duration, and age of introduction to solids. Sleep variables included bedtime, bed-sharing status (month 1), number of wakes, longest nocturnal sleep bout, nap frequency, 24 h TST (24hTST), nocturnal (nighttime) and diurnal (daytime) TST, and the proportion of nocturnal sleep time to 24hTST. To interpret GM diversity, continuous data were dichotomized by a median split, except for the bedtime variable, for which we used the bedtime recommendation of earlier than 21:00 [34]. Infant weight-for-age was converted to z-scores based on World Health Organization growth charts. The change in weight-for-age z-scores from birth to 6 months was calculated, and RWG status (>0.67 change) [35] was computed.

At 36 months, a subsample (*n* = 40) provided fecal samples for GM analysis. These samples were collected from soiled toilet paper or diapers using sterile swabs post-defecation, shipped overnight to the Clinical Research Unit at Arizona State University (ASU) and stored at −80 °C until extraction. The DNA extraction utilized the PowerSoil DNA Isolation Kit (MoBio Laboratories Ltd., Carlsbad, CA, USA). To maximize yield, the swab tip was cut into the bead tube before adding the lysing reagent. The remaining protocol followed the manufacturer’s instructions. Sample quality and concentration were verified using a high-speed microfluidic UV/VIS spectrophotometer (QIAxpert Instrument from Qiagen, Hilden, Germany), and the extracted DNA was stored at −80 °C until sent to the Biodesign Institute Genomics Core Lab at ASU for amplification and sequencing.

The 16S rRNA gene (V4 region) was amplified using 515-806R primers by polymerase chain reaction (PCR) following the Earth Microbiome Project protocol [36]. PCR amplifications for each sample were done in triplicate. DNA amplicons were pooled and quantified using the Quant-iT™ PicoGreen^®^ dsDNA Assay Kit (Invitrogen, Waltham, MA, USA). A total of 240 ng of DNA per sample was pooled and cleaned using the QIA quick PCR purification kit (Qiagen). The pooled sample was then quantified using the Illumina library Quantification Kit ABI Prism^®^ (Kapa Biosystems, Wilmington, MA, USA) and diluted to 4 nM. The sample was further denatured and diluted to a final concentration of 4 pM with 15% PhiX. The DNA library was sequenced using the MiSeq Illumina platform, version 2 module, 2 × 250 paired-end reads, following manufacturer directions. Demultiplexed sequences were imported into Quantitative Insights into Microbial Ecology version 2 [37]. DADA2 was used to remove chimeras and sequences with a quality score of <30. To further refine the sequence data for analysis, rarefaction curves were evaluated, and a sampling depth of 9097 sequences was established. This depth ultimately led to the exclusion of data from four samples, resulting in a final sample size of 36 toddler fecal microbiomes. Taxonomic assignment and phylogenetic trees were generated based on the SILVA 138 database [38].

Statistical Analysis: Demographic and predictor variable data were described and analyzed with SPSS (version 28.0, IBM Corp., Armonk, NY, USA). Dichotomous variables were analyzed using the chi-squared test (two-sided) and continuous variables were inspected for normality and analyzed by two-sided Mann–Whitney and *t*-tests for non-normal and normal distributions, respectively. Predictor variables at each time point (i.e., 1, 6, and 12 months) were analyzed. A *p*-value < 0.05 was considered statistically significant. We evaluated differences in alpha diversity (i.e., within-subject variability in microbial diversity as measured by Shannon, Pielou’s Evenness, and Faith’s PD) by feeding, sleep, and growth variables using Kruskal–Wallis tests. Beta diversity distance matrices (unweighted and weighted UniFrac, Bray–Curtis, and Jaccard) were constructed, and differences in between-subject diversity by the predictor variables were calculated using PERMANOVA analysis. Variables that predicted significant differences in diversity were evaluated using Linear Discriminant Analysis Effect Size (LEfSe) analysis with an alpha = 0.05 and an effect size = 3.0 for both genus and family to discover potential microbial biomarkers [39]. If multiple relevant predictors of phylogenetic diversity (Faith PD, examining phylogeny within-sample) across feeding, sleep, and growth were found at a given age, then forward stepwise regression models were computed to confirm whether the associations persisted in the presence of each other using JMP Pro 16 software. Whenever possible, the corresponding continuous variables were used. The final model was selected based on the lowest Bayesian Information Criterion (BIC). Similarly, a Permutational Multivariate Analysis of Variance (Adonis) was conducted to determine the variance attribution of the same variables using the unweighted UniFrac matrix (accounting for phylogenetic between-sample differences).

## 3. Results

There were no significant differences in demographic, anthropometric, or perinatal characteristics/behaviors between the 36-infant subsample and the larger cohort (Table 1 and Appendix A). Toddler GM diversity analysis did not differ by delivery mode, sex, or based on the educational intervention of the parental study.

### 3.1. Feeding Practices

Feeding variables used in this study are listed in Table 2. The GM structure between samples at three years of age differed between infants who received any breastmilk and those exclusively formula fed at 1 month of age (Bray–Curtis, pseudo-F = 1.805, *p* = 0.018). This difference was not observed when evaluating feeding mode at 6 and 12 months of age. Toddler GM community structure differed between those introduced to solid foods before the recommended time of 6 months and those introduced later (Bray–Curtis, pseudo-F = 1.651, *p* = 0.044).

### 3.2. Sleeping Patterns

Sleep variables as predictors of microbial diversity are displayed in Table 3.

Greater microbial richness was observed in toddlerhood among children who experienced bed-sharing with their caregiver(s) at 1 month of age (Shannon H = 4.335, *p* = 0.037). Additionally, infants with a lower proportion of night to 24hTST at 1 month had a richer (Shannon H = 4.395, *p* = 0.036) and more even (Evenness H = 4.102, *p* = 0.043) GM in toddlerhood. Moreover, the proportion of night to 24hTST was associated with microbial community structure differences that were driven by abundance (Bray–Curtis, pseudo-F = 1.421, *p* = 0.035) and phylogeny (Unweighted Unifrac, pseudo-F = 1.428, *p* = 0.030). Similarly, 6-month-old infants who had a lower proportion of night to 24hTST had a higher within-sample microbial richness (OTU, H = 5.559, *p* = 0.018) and phylogenetic diversity in toddlerhood (Figure 1B, Faith PD H = 11.676, *p* = 0.001). Differences by proportion of nighttime sleep in 6-month-olds were also seen in their GM community structure (between-sample beta diversity, Figure 2), and diurnal TST demonstrated similar findings (Appendix A). At 6 months of age, infants who slept more than 13 h throughout the 24 h day had higher phylogenetic diversity in toddlerhood than infants who slept 13 h or less (Figure 1A, Faith PD H = 4.550, *p* = 0.033). Twelve-month-olds with bedtimes earlier than 21:00 had a richer (Faith PD H = 3.854, *p* = 0.050) GM in toddlerhood than infants with bedtimes at 21:00 or later. The overall community structures also differed at three years of age by bedtime classification (Weighted Unifrac, pseudo-F = 3.018, *p* = 0.034). No other sleep variables were associated with GM diversity.

### 3.3. Rapid Weight Gain (RWG)

Infants that experienced RWG from birth to 6 months did not present significant GM differences in toddlerhood for any of the measured diversity metrics. However, phylogenetic diversity was slightly greater among toddlers who did not experience RWG in infancy (Faith PD H = 3.6327, *p* = 0.057).

### 3.4. Feeding, Sleep, and Growth as Predictors of Phylogenetic Diversity

Given that significant and relevant associations for feeding, sleep, and growth with Faith PD were found at 6 months of age, a forward stepwise regression model was constructed using the age of starting solids, the proportion of night to 24hTST at 6 months, and the change of weight-for-age z-scores from birth to 6 months (RWG). The final model (R = 20.9, *p* = 0.008) retained both the proportion of night to 24hTST and RWG (Table 4). The multivariate analysis (Adonis) of between-sample diversity using the unweighted Unifrac metric confirmed that the proportion of night to 24hTST at 6 months was the only significant predictor of overall GM community structure (R^2^ = 8.12%, *p* = 0.001).

### 3.5. Microbial Taxa Differentiated by Feeding, Sleep, and Growth Variables

The list of the genera and families associated with the predictors is in Table 5. Infants that were not breastfed at 1 month had a greater abundance of *Rothia* as toddlers. Introduction to solids before 6 months was associated with a higher abundance of *Monoglobus*. A higher abundance of the family *Tissierellales* was observed in toddlers who slept more than 13 h at 6 months, while *Akkermansia* and *Bifidobacterium* were more prominent in toddlers who slept less than 13 h as 6-month-old infants. Genera associated with both a lower proportion of night-to-24hTST and greater daytime TST at 6 months were: *Campylobacter*, *Christensenellaceae_r_7_group*, *Enterococcus*, *Anaerococcus, and Fenollaria.* On the other hand, *Bacteroides* was associated with more nighttime sleep and less daytime TST. Significant differences in microbial abundance were not observed by RWG status.

## 4. Discussion

In this study, we explored the relationships between feeding practices, sleeping patterns, and growth measured at three different timepoints during infancy and the GM composition in toddlerhood. We found early breastmilk consumption, the age of starting solid foods, bed-sharing, bedtime, 24hTST, and the proportion of night to 24hTST differentiated the toddler’s GM. We also identified specific GM taxa associated with each behavior.

Our findings suggest that toddlers who received any breastmilk at 1 month of age had a different GM community structure when compared to those not receiving any breastmilk. This microbial imprint of early breastfeeding was not observed when evaluating feeding practices at 6 and 12 months of age. This may be explained by the introduction of solid foods, a milestone that profoundly influences GM maturation [40], with 86.1% of our infants being introduced to solids prior to 6 months. Unexpectedly, the LEfSe analysis did not identify *Bifidobacterium* as a breastfeeding biomarker, potentially due to this cohort’s early cessation of breastfeeding prior to the 2-year recommendation [41] and given that this population gradually decreases breastfeeding as soon as infants are introduced to solids [11]. Nonetheless, *Rothia* was associated with infants who did not breastfeed at 1 month, which aligns with previous work linking this microbe to formula consumption [42], making this taxa a potential biomarker for formula feeding practices. *Monoglobus* has been associated with the degradation of pectin, a carbohydrate present in fruits [43], and in this study, it was found in the GM of toddlers introduced to solids before 6 months. Early feeding is critical for childhood growth and development [44]; specifically, breastfeeding provides access to a variety of unique compounds that shape the microbiome by seeding the infant’s gut, thereby creating a lasting impact into toddlerhood [45,46]. Collectively, our findings highlight the lasting impact of early feeding practices.

The proportion of night to 24hTST at 1 and 6 months was the sleep variable with the most prominent association to the toddlerhood GM; infants with a lower proportion of nighttime sleep had greater microbial richness and a unique GM community structure driven by abundance and phylogenetic differences when compared to toddlers with a higher proportion of nighttime sleep. Moreover, this association persisted when including infant growth and feeding variables in a stepwise regression analysis. Our analysis builds on previous findings suggesting that a decreased proportion of night to 24hTST among three-month-olds is associated with lower microbial richness [27]. In contrast, our data suggest that a lower proportion of night to 24hTST at 1 and 6 months is related to higher alpha diversity by the time children reach toddlerhood. One plausible reason for this could be that alpha diversity increases with age, and different factors affect the rate and magnitude of change. More work is needed to understand rates of change and associated health outcomes, given that current data suggest a faster rate of maturation of the GM may not necessarily be optimal [40,47].

Several findings from this study add to the GM literature in relation to infant sleep recommendations. Although the American Academy of Pediatrics (AAP) [48] does not recommend bed-sharing with an infant for safety reasons, participants reported engaging in this behavior. Interestingly, bed-sharing at 1 month of age was associated with a richer toddler GM. This observation aligns with the hypothesis suggesting that this practice can facilitate the transfer of microbes through skin contact, although other variables, such as breastfeeding, may also play a role in this transfer [49]. We also found that earlier bedtimes at 12 months were associated with a more diverse GM in toddlerhood, adding a timing variable to the complex interactions between sleep hygiene and the GM. While the AAP and the National Sleep Foundation encourage 12–16 h of daily sleep for infants aged 0–12 months, there are no standardized recommendations for a bedtime range [34]. In adults, there is evidence that a temporal shift in bedtime is accompanied by temporal alterations in GM functionality [50]. This dynamic is worthy of further exploration in early childhood, as similar associations may exist.

In our study, the genus *Bacteroides* was positively associated with a higher proportion of nighttime sleep. Previous research has linked this genus to short sleep duration in adults [51]; however, its role in health and disease is species-dependent [51,52,53]. Genera differentiated by sleeping patterns (Table 5) have been associated with both positive and negative health outcomes [13,53,54,55,56,57], and their roles in early childhood remain unclear. Nonetheless, the presence or absence of this bacteria could offer critical insights into the long-term health implications of sleep patterns, making it a potential microbial biomarker for future sleep interventions in infants.

Our analysis shows that infants experiencing RWG at 6 months had a lower GM phylogenetic diversity in toddlerhood, although this difference was found to be non-significant. Nonetheless, this trend is noteworthy since it adds to a previous analysis that used the same cohort and demonstrated an association between RWG at 6 months and increased odds of being overweight in toddlerhood [7]. This association has also been observed by others [58] and underscores the role of early weight-gain patterns in long-term health. Our study’s indication of a lower GM diversity being associated with RWG could be linked to the known adverse health outcomes of overweight and obesity that have been previously associated with a decreased GM diversity [53,56]. Given that GM composition seems to be established by 3 years of age [14,59], our findings highlight a crucial window for intervention.

This work has several strengths. Our sample was exclusively infants born to low-income Latinas with obesity, a demographic overlooked in GM research with a higher risk for excessive gestational weight gain and RWG of their offspring [30,31]. Additionally, our data collection method allowed us to connect feeding and sleep behaviors in infancy with GM composition at three years of age. Much of the GM literature remains cross-sectional, and this study offers ideas for longitudinal studies. Lastly, the subsample used in this study was representative of the larger cohort; however, larger sample sizes may be required in infant sleep and microbiome studies given that sleep in infancy has high variability [60]. Limitations of this study include having a small sample size for this exploratory analysis and the potential bias from obtaining the feeding and sleep metrics through retrospective questionnaires. Additionally, the added variability that may have arisen from collecting GM samples from two sources, toilet paper and diapers, could have influenced the results.

## 5. Conclusions

Feeding practices, sleep patterns, and growth trajectories in infancy play a role in shaping the GM, with their effects lasting through three years of age. We demonstrated that breastfeeding at 1 month and starting solids before the age of 6 months were associated with differences in GM diversity and community structure in toddlerhood. A lower proportion of daily nighttime sleep at both 1 and 6 months of age was associated with greater GM diversity in toddlerhood. Finally, the change in weight-for-age z-scores emerged as a potential predictor of phylogenetic diversity, but larger samples are needed to confirm this finding. These findings may have implications for clinical recommendations, but longitudinal work is needed to understand temporal changes in sleep and microbial taxa development throughout the first year after birth and how sleep and feeding behaviors interact with the GM to influence obesity risk in toddlerhood and childhood.

## Figures and Tables

**Figure 1 microorganisms-12-00549-f001:**
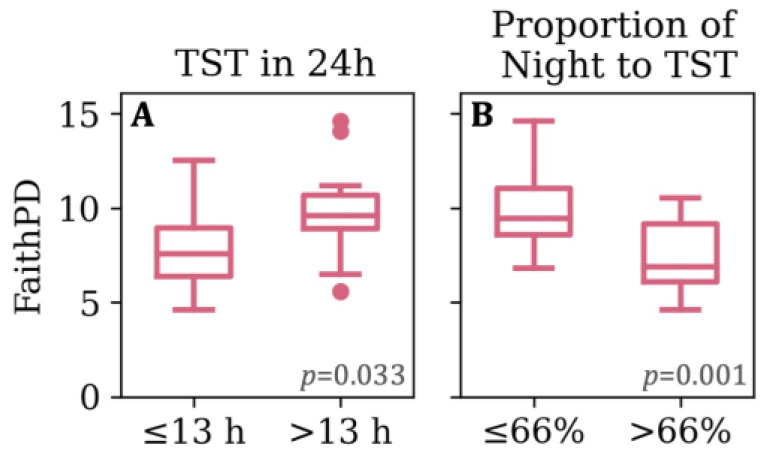
Alpha diversity (Faith PD) of toddlers at 3 years of age by (**A**) total sleep time in a 24 h period (24hTST); and (**B**) proportion of nighttime sleep relative to 24hTST reported by mothers at 6 months of age.

**Figure 2 microorganisms-12-00549-f002:**
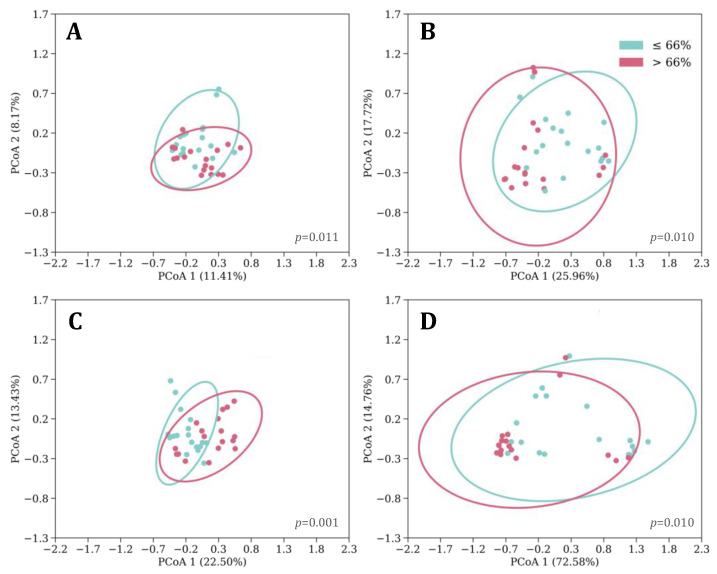
Beta diversity plots for infants with >66% (pink) and ≤66% (blue) proportions of their total 24 h sleep at night at 6 months with 95% confidence ellipses and PERMANOVA *p*-values using the following diversity metrics: (**A**) Jaccard; (**B**) Bray–Curtis; (**C**) unweighted Unifrac; (**D**) weighted Unifrac.

**Table 1 microorganisms-12-00549-t001:** Maternal and infant characteristics by the original and analytic subsample.

Variable	*n* = 126 Cohort	*n* = 36 Subsample	Significance (*p* Value)
Infant sex; female, *n* (%) ^a^	55 (43.7)	16 (44.4)	0.92
Birth mode, n (%) a			0.48
- Vaginal	69 (54.8)	18 (50.0)	
- Cesarean-section	48 (38.1)	15 (41.7)	
- No response	9 (7.1)	3 (8.3)	
Infant birth weight-for-age Z score, *M* (*SD*) ^b^	0.42 (0.9)	0.41 (1.0)	0.95
Rapid weight gain ^d^ at 6 months, *n* (%) ^a^	45 (35.7)	11 (30.6)	0.50
Maternal education, n (%) a			0.56
- <High school graduate	64 (50.8)	21 (58.3)	
- ≥High school graduate	58 (46.0)	14 (38.9)	
- Other	4 (3.2)	1 (2.8)	
Maternal parity, n (%) a			0.17
- 0	32 (25.4)	5 (13.9)	
- 1	35 (27.8)	9 (25.0)	
- 2	27 (21.4)	9 (25.0)	
- ≥3	32 (25.5)	13 (36.1)	
Maternal prenatal BMI (kg/m^2^), *M* (SD) ^b^	35.9 (5.2)	36.6 (6.0)	0.36
Gestational weight gain (lb), *M* (*SD*) ^c^	25.3 (11.2)	25.0 (13.9)	0.86
Breastfeeding duration by 12-month follow-up (months), *M* (*SD*) ^c^	4.4 (4.7)	5.1 (5.0)	0.38
Exclusive breastfeeding duration (months), *M* (*SD*) ^c^	2.3 (4.1)	3.3 (4.5)	0.48
Age of introduction to solids (months), *M* (*SD*) ^c^	4.9 (1.1)	4.7 (1.0)	0.08

^a^ Chi-square test. ^b^ *t*-test. ^c^ Mann–Whitney U-test. ^d^ Rapid weight gain defined by an increase of >0.67 in weight-by-age z-scores from birth.

**Table 2 microorganisms-12-00549-t002:** Feeding characteristics during infancy (*n* = 36).

Variable	Month		
	1	6	12
Any breastfeeding, n (%)			
- Yes	25 (69.4)	15 (41.7)	8 (22.2)
- No	11 (30.6)	21 (58.3)	28 (77.8)
Milk exclusivity, n (%)			
- Exclusively breastmilk	16 (44.4)	10 (27.8)	2 (5.6)
- Not exclusive to breastmilk	20 (55.6)	26 (72.2)	34 (94.4)
Started solids before time point, n (%)			
- Yes	0 (0.0)	31 (86.1)	36 (100.0)
- No	36 (100.0)	5 (13.9)	0 (0.0)

**Table 3 microorganisms-12-00549-t003:** Sleep characteristics (*n* = 36) and their dichotomous variable construction by median splits.

Variable			Month			
	1		6		12	
	M (SD) [Range]	DV (*n*)	M (SD) [Range]	DV (*n*)	M (SD) [Range]	DV (*n*)
Bed-sharing, *n* (%)	8 (22.0)	Yes (8) No (27)				
Bedtime (hh:mm) ^a^	21:45 (0:59) [20:00, 23:00]	<21:00 (7) ≥21:00 (28)	21:37 (0:53) [20:00, 00:00]	<21:00 (4) ≥21:00 (32)	21:45 (0:57) [19:00, 00:30]	<21:00 (4) ≥21:00 (32)
Number of wakes	2.8 (1.4) [0, 8]	≤3 (27) >3 (8)	1.1 (1.0) [0, 3]	≤1 (21) >1 (15)	1.1 (1.0) [0, 3]	≤1 (25) >1 (11)
Longest nocturnal sleep bout (h)	4.9 (1.9) [3, 10]	≤4 (22) >4 (13)	7.2 (2.0) [4, 11]	≤7 (19) >7 (17)	8.6 (2.3) [4, 12]	≤9 (21) >9 (15)
Nocturnal Sleep (h)	9.7 (1.3) [7, 12]	≤10 (27) >10 (8)	8.9 (1.5) [6, 11]	≤9 (19) >9 (17)	9.8 (0.9) [8, 12]	≤10 (30) >10 (6)
Nap frequency	4.0 (1.3) [1, 8]	≤4 (27) >4 (8)	2.9 (0.9) [2, 5]	≤3 (25) >3 (11)	2.0 (0.9) [1, 4]	≤2 (27) >2 (9)
Daytime Sleep (h)	7.8 (1.1) [5.0, 9.0]	≤8 (26) >8 (9)	4.6 (2.0) [0.5, 8.0]	≤5 (25) >5 (11)	2.9 (1.5) [0.5, 6.0]	≤3 (22) >3 (14)
24hTST	17.5 (1.9) [13, 21]	≤18 (25) >18 (10)	13.6 (2.1) [8.7, 19.0]	≤13 (20) >13 (16)	12.8 (1.7) [9, 16]	≤13 (22) >13 (14)
Proportion of Night to 24hTST	0.6 (0.0) [0.5, 0.7]	≤0.56 (24) >0.56 (11)	0.7 (0.1) [0.5, 1.0]	≤0.66 (18) >0.66 (18)	0.8 (0.1) [0.6, 1.0]	≤0.77 (20) >0.77 (16)

^a^ Bedtime was dichotomized by the recommended bedtime of 21:00 or earlier [34]. The median bedtimes at 1, 6, and 12 months were 22:00, 21:45, and 22:00, respectively. DV: dichotomous variable from median splits.

**Table 4 microorganisms-12-00549-t004:** Final multivariate model of within-sample gut microbial diversity (Faith PD) following stepwise regression of feeding, sleep, and growth predictors at 6 months of age.

	Model	
Rsquare Adj	20.9	
F Ratio (df) ^a^	5.62 (2, 33)	
*p*-value	0.008	
	Parameter (SE)	*p*-value
Intercept	14.18 (1.99)	<0.001
Proportion Night to 24hTST at 6 months	−8.07 (2.94)	0.010
ΔWeight-for-age z-scores from birth to 6 months	−0.87 (0.44)	0.056

^a^ (df from regression, df from error).

**Table 5 microorganisms-12-00549-t005:** Results of differential abundance analysis to identify microbial taxa with an LDA score higher than 3.0 for genus and corresponding family by feeding and sleep categories.

Family	Genus	Feeding or Sleep Variable	Association	Age of Infant When Assessed	LDA Score	*p*-Value
*Actinomycetaceae*	*Varibaculum*	Proportion of Night to 24hTST ^a^	66% or less	6	3.585	0.037
*Akkermansiaceae*	*Akkermansia*	24hTST	13 h or less	6	4.255	0.046
*Bacillaceae*	*Bacillus*	Daytime Sleep	More than 5 h	6	3.367	0.007
*Bacteroidaceae*	*Bacteroides*	Daytime Sleep	5 h or less	6	4.802	0.027
		Proportion of Night to 24hTST	More than 66%	6	4.911	0.002
*Bifidobacteriaceae*	*Bifidobacterium*	Bedtime	21:00 and after	12	4.156	0.037
		24hTST	13 h or less	6	4.175	0.032
	*Gardnerella*	Bedtime	Before 21:00	12	3.132	0.005
*Campylobacteraceae*	*Campylobacter*	Proportion of Night to 24hTST	66% or less	6	3.246	0.018
		Daytime Sleep	More than 5 h	6	3.183	0.011
*Christensenellaceae*	*Christensenellaceae_R_7_group*	Proportion of Night to 24hTST	66% or less	6	3.232	0.021
		Daytime Sleep	More than 5 h	6	3.190	0.034
*Clostridia_UCG_014*	*Clostridia_UCG_014*	Bed-sharing	Bed-sharing	1	3.844	0.027
		Bedtime	Before 21:00	12	3.989	0.011
		Proportion of Night to 24hTST	66% or less	6	3.567	0.035
*Clostridia_vadinBB60_group*	*Clostridia_vadinBB60_group*	Bedtime	Before 21:00	12	3.120	0.005
*Enterobacteriaceae*	*Escherichia_Shigella*	Proportion of Night to 24hTST	More than 56%	1	4.761	0.043
		Daytime Sleep	More than 5 h	6	4.662	0.004
*Enterococcaceae*	*Enterococcus*	Proportion of Night to 24hTST	66% or less	6	3.204	0.016
		Daytime Sleep	More than 5 h	6	3.263	0.027
*Erysipelotrichaceae*	*Holdemanella*	Proportion of Night to 24hTST	66% or less	6	3.416	0.007
	*Solobacterium*	Proportion of Night to 24hTST	66% or less	6	3.660	0.037
*Gastranaerophilales*	*Gastranaerophilales*	Bedtime	Before 21:00	12	3.732	0.000
*Hafniaceae*	*Hafnia_Obesumbacterium*	Bedtime	Before 21:00	12	3.152	0.005
*Listeriaceae*	*Listeria*	Bedtime	Before 21:00	12	3.159	0.005
*Methanobacteriaceae*	*Methanobrevibacter*	Bedtime	Before 21:00	12	3.233	0.005
*Micrococcaceae*	*Rothia*	Any breastfeeding	No	1	3.949	0.034
*Monoglobaceae*	*Monoglobus*	Age of Starting Solids	Before 6 months	6	3.421	0.026
*Muribaculaceae*	*Muribaculaceae*	24hTST	More than 13 h	6	3.004	0.046
*Oscillospiraceae*	*UCG_002*	Bedtime	Before 21:00	12	4.050	0.038
	*UCG_005*	Bedtime	Before 21:00	12	3.135	0.014
*Porphyromonadaceae*	*Porphyromonas*	Proportion of Night to 24hTST	66% or less	6	3.234	0.001
		24hTST	More than 13 h	6	3.047	0.011
*Prevotellaceae*	*Alloprevotella*	Bedtime	Before 21:00	12	4.005	0.002
	*Prevotella*	Bedtime	Before 21:00	12	4.971	0.027
		Proportion of Night to 24hTST	66% or less	6	4.697	0.002
	*Prevotellaceae_UCG_003*	Bedtime	Before 21:00	12	3.353	0.005
*Pseudoalteromonadaceae*	*Pseudoalteromonas*	Bedtime	Before 21:00	12	3.243	0.005
*RF39*	*RF39*	Proportion of Night to 24hTST	56% or less	1	3.473	0.014
		Bedtime	Before 21:00	12	3.345	0.022
		24hTST	More than 13 h	6	3.232	0.037
*Sutterellaceae*	*Sutterella*	Proportion of Night to 24hTST	56% or less	1	3.792	0.015
*Tissierellales*	*Anaerococcus*	Proportion of Night to 24hTST	66% or less	6	3.345	0.008
		24hTST	More than 13 h	6	3.313	0.002
		Daytime Sleep	More than 5 h	6	3.474	0.000
	*Ezakiella*	Proportion of Night to 24hTST	66% or less	6	3.174	0.008
		24hTST	More than 13 h	6	3.299	0.029
	*Fenollaria*	Proportion of Night to 24hTST	66% or less	6	3.428	0.033
		24hTST	More than 13 h	6	3.287	0.021
		Daytime Sleep	More than 5 h	6	3.484	0.002
	*Finegoldia*	24hTST	More than 13 h	6	3.158	0.046
		Daytime Sleep	More than 5 h	6	3.224	0.007
	*Gallicola*	Proportion of Night to 24hTST	66% or less	6	4.286	0.018
		24hTST	More than 13 h	6	3.738	0.008
	*Murdochiella*	Proportion of Night to 24hTST	66% or less	6	4.079	0.037
		24hTST	More than 13 h	6	3.436	0.020
	*Peptoniphilus*	24hTST	More than 13 h	6	3.365	0.042
		Daytime Sleep	More than 5 h	6	3.514	0.001
*Veillonellaceae*	*Dialister*	Proportion of Night to 24hTST	More than 56%	1	3.348	0.032
		Daytime Sleep	More than 5 h	6	3.921	0.007
*Victivallaceae*	*Victivallis*	Bedtime	Before 21:00	12	3.012	0.003

^a^ 24 h total sleep time (24hTST).

## Data Availability

The data presented in this study are available on request from the corresponding author.

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
