# Peer review of "Associations of Infant Feeding, Sleep, and Weight Gain with the Toddler Gut Microbiome"

_microorganisms, 2024, doi:10.3390/microorganisms12030549_

Round 1
Reviewer 1 Report
Comments and Suggestions for Authors
The article "Infant feeding, sleep, and rapid weight gain are associated with the gut microbiome at 3 years among infants born to low-income Latina women with obesity" aims to evaluate feeding practices, sleep patterns, and rate of growth during infancy with their impact on gut microbiome at age three years digestion. Concerning the structure and the form of the manuscript, it would be recommended that the guidelines for authors be read and the instructions respected carefully. The article's structure is unclear, some ideas are repeated, and the analytical methods are not chronologically exposed. The English written is not satisfactory, and I recommend it be enhanced by a native person who speaks native English. Before publication, Valuable knowledge must be added to the materials, results, and discussion chapter.
1. The title is not representative of the conducted study. The title of your manuscript should be concise, specific, and relevant.
2. The abstract should be a maximum of about 200 words.
3. I recommend checking the citations in the text. Consider checking the journal guidelines.
4. “Materials and Methods”
- Clarify the rationale behind selecting pregnant Latina women with low income and obesity as the study population. Discuss any specific characteristics of this population that make it relevant to the research question.
- Consider specifying unique ethical considerations related to the study population or the nature of the research.
- Please indicate whether collecting fecal samples from toilet paper or diapers might result in any sample variability. Please clarify potential considerations related to sample contamination or differences in collection methods.
5. The results and discussions from the manuscript need to be revised. The discussions present too general data. How can the authors explain the results they obtained? Most of the discussions have been superficially discussed. The authors should focus precisely on the results obtained and discuss them more deeply with specific original research from the domain.
- How might these findings contribute to readers' understanding of early childhood development?
- How may the presence or absence of these bacteria influence readers' comprehension of the effects of early feeding practices on the gut microbiome?
- Line 259 “ Our findings expand….” Compare your results with the mentioned study.
6. References- Please check the format of the references. I recommend checking the guide for reference format.
Reviewer 2 Report
Comments and Suggestions for Authors
The manuscript titled “Infant feeding, sleep and rapid weight gain are associated with the gut microbiome at 3 years among infants born to low-income Latina women with obesity”, is an interesting study on the association between food source, sleep pattern, sleep length of infants and their gut microbiome composition at the age of three. Please find my comments as follows:
- Title: Revise the title to improve its coherence and rhythm!
- Add an explanation to the introduction explaining the composition of the gut microbiome, pointing to "What are the most common gut microorganisms (adults and infants)?", and "How can sleep and breastfeeding influence gut microbiome composition?"
- Line 48-49: what do you mean about the exclusivity? do you mean formula ingredients "formula supplementation"?
- Line 53-54: Clarify what the alfa diversity is!
- Line 57: under study not understudied!
- Line 58: It is worth it to explain how various studies (methods and approaches) demonstrated that different factors such as sleep, formula ingredients, etc. could affect the GM composition.
Line 61: What is optional GM health?
Comments on the Quality of English LanguageMinor editing of the English language is needed.
Round 2
Reviewer 1 Report
Comments and Suggestions for Authors
The authors have adequately addressed all reviewer's comments.
The paper is ready to be published.